# miRNome and Proteome Profiling of Human Keratinocytes and Adipose Derived Stem Cells Proposed miRNA-Mediated Regulations of Epidermal Growth Factor and Interleukin 1-Alpha

**DOI:** 10.3390/ijms24054956

**Published:** 2023-03-04

**Authors:** Hady Shahin, Sallam Abdallah, Jyotirmoy Das, Weihai He, Ibrahim El-Serafi, Ingrid Steinvall, Folke Sjöberg, Moustafa Elmasry, Ahmed T. El-Serafi

**Affiliations:** 1Department of Hand Surgery, Plastic Surgery and Burns, Linköping University, 58185 Linköping, Sweden; 2The Department of Biomedical and Clinical Sciences, Linköping University, 58183 Linköping, Sweden; 3Faculty of Biotechnology, Modern Sciences and Arts University, Cario 12585, Egypt; 4Bioinformatics, Core Facility, Division of Cell Biology, Department of Biomedical and Clinical Sciences, Faculty of Medicine and Health Sciences, Linköping University, 58183 Linköping, Sweden; 5Clinical Genomics Linköping, SciLife Laboratory, Department of Biomedical and Clinical Sciences, Faculty of Medicine and Health Sciences, Linköping University, 58183 Linköping, Sweden; 6Basic Medical Sciences Department, College of Medicine, Ajman University, Ajman P.O. Box 346, United Arab Emirates; 7Department of Biochemistry, Faculty of Medicine, Port-Said University, Port Fouad City 42526, Egypt; 8Medical Biochemistry and Molecular Biology Department, Faculty of Medicine, Suez Canal University, Ismailia 41522, Egypt

**Keywords:** keratinocytes, adipose-derived stem cells, direct co-culture, miRNA, proteome, epidermal growth factor, interleukin 1 alpha, stem cell differentiation

## Abstract

Wound healing is regulated by complex crosstalk between keratinocytes and other cell types, including stem cells. In this study, a 7-day direct co-culture model of human keratinocytes and adipose-derived stem cells (ADSCs) was proposed to study the interaction between the two cell types, in order to identify regulators of ADSCs differentiation toward the epidermal lineage. As major mediators of cell communication, miRNome and proteome profiles in cell lysates of cultured human keratinocytes and ADSCs were explored through experimental and computational analyses. GeneChip^®^ miRNA microarray, identified 378 differentially expressed miRNAs; of these, 114 miRNAs were upregulated and 264 miRNAs were downregulated in keratinocytes. According to miRNA target prediction databases and the Expression Atlas database, 109 skin-related genes were obtained. Pathway enrichment analysis revealed 14 pathways including vesicle-mediated transport, signaling by interleukin, and others. Proteome profiling showed a significant upregulation of the epidermal growth factor (EGF) and Interleukin 1-alpha (IL-1α) compared to ADSCs. Integrated analysis through cross-matching the differentially expressed miRNA and proteins suggested two potential pathways for regulations of epidermal differentiation; the first is EGF-based through the downregulation of miR-485-5p and miR-6765-5p and/or the upregulation of miR-4459. The second is mediated by IL-1α overexpression through four isomers of miR-30-5p and miR-181a-5p.

## 1. Introduction

Skin is a self-renewing organ that covers the entire surface area of the body. It forms an anatomical guard that works as a barrier from the outer environment. Skin broadly consists of 3 layers: epidermis, dermis, and subcutaneous adipose tissue [1,2]. Because of the complex organization, wound healing involves orchestrated synergy between different skin cells with interplay between a plethora of signaling chemokines, growth factors, and cytokines. The healing process starts with the formation of a fibrin clot, followed by the recruitment of inflammatory cells. Granulation tissue then starts to form alongside angiogenesis. Re-epithelialization takes place with the recruitment of proliferating fibroblasts and migrating keratinocytes causing the dermis to contract [3,4]. Epithelialization is an integral component of wound healing used as a decisive factor of its success. Impaired epithelialization is a characterizing feature of chronic wounds [5]. Keratinocytes, the predominant cellular constituent of the epidermis, play a central role in restoring the epidermis after injury as epithelization is largely mediated by local keratinocytes at the wound edges and by epithelial stem cells in the hair follicles or sweat glands [3,4].

Adipose-derived stem cells (ADSCs), known for their plasticity and low immunogenicity, can be isolated by minimally invasive procedures from subcutaneous adipose tissue with a remarkable yield of ADSCs [6]. These cells have shown potential to differentiate into cell types from different lineages, including osteogenic, chondrogenic, neural, cardiomyogenic, hepatic, endocrinal, as well as epithelial lineages [7,8,9,10]. Additionally, the plasticity and immunomodulatory capacity of ADSCs could increase the chances for the success of cellular transplantation [11]. Establishing sustainable cultures of epidermal cells while eliminating the progressive loss of epidermal stem cell population can be a challenging process due to rapid clonal conversion [12]. As a result, in vitro differentiation protocols were considered to convert ADSCs into epidermal-like cells [13]. Several reports tried to optimize epidermal differentiation protocol with organic and non-organic culture media additives [14,15]. Unfortunately, the effectiveness, reproducibility, and compatibility with the regulations for clinical use are always challenging.

Cross-talk within the epidermal niche between keratinocytes, fibroblasts, stem cells, endothelium, immune cells, and other cell types is vital for successful cutaneous wound repair [5]. Keratinocytes release signaling molecules that act in an autocrine and paracrine manner to stimulate the epithelialization through modulating the proliferative and migratory pattern of the surrounding cells. Multiple signaling molecules, including cytokines, chemokines, growth factors, integrins, and coding and non-coding RNAs function as active mediators of cell–cell communication and are crucial for its effectiveness [5,16]. Furthermore, dermal adipocytes and ADSCs play important roles during skin repair via endocrine secretions and signaling [17,18,19,20,21]. Extracellular vesicles (EVs) can promote early stages of healing through pleiotropic effects, including enhancing fibroblasts migratory and proliferative capacity, as well as inducing collagen deposition. EVs derived from ADSCs have been reported to accelerate murine cutaneous wound healing when applied through local and intravenous injections [20]. MicroRNAs (miRNAs) are small non-coding RNAs that are a major constituent of the EVs cargo; miRNAs can regulate gene expression in neighboring cells when released extracellularly, either freely or within Evs. Additionally, miRNAs play a role in all major cellular processes, including metabolism, cell proliferation, differentiation, and apoptosis [2,22]. A single miRNA can regulate several mRNAs at the post-transcriptional level, but several miRNAs can conversely bind to a single gene and cooperatively fine-tune its expression, which attests to the complexity of miRNA-mediated gene regulation [22,23]. Epigenetically, miRNAs can activate/repress gene expression by controlling the rate of transcription or translation [24,25]. Additionally, miRNAs play a central role in epidermal development where various miRNAs have been detected throughout skin cell lineages during embryonic skin morphogenesis [26]. Studies involving conditional depletion of either of the major regulators of miRNAs biogenesis, Dicer and Drosha, in murine epidermis showed the loss of barrier function, aberrations in hair follicle growth, and impaired epidermal differentiation [26,27]. Various studies identified a multitude of miRNAs and their role in regulating key epidermal developmental processes by creating feedback loops that modulate the proliferation, differentiation, and the migration of epidermal cells in both normal and disease conditions [27,28,29,30]. Additionally, miRNAs are involved in the overlapping phases of cutaneous wound healing, including the inflammatory phase, and as regulators of angiogenesis during re-epithelialization [2,31]. In the context of stem cells, miRNAs are involved in the regulatory pathways modulating the differentiation process of ADSCs into various cell lineages including osteogenic, chondrogenic, neuronal and adipogenic differentiation [32,33,34].

In this study, the effect of a direct co-culture model on stem cell differentiation is reported. The model consists of human keratinocytes and ADSCs co-cultivated at equal numbers. Additionally, this study aims to characterize the differential repertoire of miRNAs and proteins in the lysate of cultured human keratinocytes and ADSCs. A combined approach of computational and experimental analyses was employed (Figure 1) in order to establish a miRNome-proteome axis with a focus on potential regulations that can alter the fate of ADSCs to differentiate into keratinocyte-like cells.

## 2. Results

### 2.1. Direct Co-Culture Can Enhance the Commitment of ADSCs towards the Epidermal Lineage

HaCaT, the epidermal cell line, was considered a positive control (100% group) and ASC52, the immortalized adipose-derived stem cell line, was considered a negative control (0% group). The study group consisted of a mixture of an equal number of both cell lines (50% group) as a direct co-culture system, as shown in Figure 2a–c.

After 7 days in culture, gene expression analysis indicated that direct co-cultures of HaCaT with ASC52 expressed higher levels of the transcription factor p63 and the early epithelial marker KRT18 (0.9 folds) while both markers could not be detected in the 0% group (*p*-value = 0.0004 and 0.01 respectively). The expression of both markers in the co-culture (50% group) was similar to f HaCaT cells (100% group) with a *p*-value = 0.51 and 0.67, respectively (Figure 2d,e). Similar up-regulation pattern was shown for the basal-specific epidermal marker KRT5 (*p*-value = 0.0004; Figure 2f). Interestingly the co-culture group showed a trend of upregulation of the basal-specific epidermal marker KRT14 with 0.4 folds over the 100% group (*p*-value = 0.051; Figure 2g) while it kept the same upregulation pattern in comparison to the 0% group (*p*-value = 0.0005; Figure 2g). On the contrary, cells at day 7 in a co-culture did not express the late epidermal differentiation markers KRT1 and 10 compared with that of the ASC52 cells in the 0% group (*p*-value = 0.1883 and 0.1841, respectively; Figure 2h,i).

To confirm our initial findings, the expression of the selected epidermal-specific differentiation markers was evaluated with immunocytochemistry (ICC) after 7 days in culture. Monocultures of HaCaT (100% group) and ASC52 (0% group) were used as positive and negative controls, respectively. ASC52 cells in the 0% group exhibited no expression for KRT18 or 5 (Figure 2j,n) and faint expression of KRT14 (Figure 2r). Interestingly, in the 50% group, some of the ASC52 cells, with their distinct “spindle-like” morphology, were clearly expressing the three studied markers, especially the cells in close proximity to HaCaT colonies (Figure 2k,o,s). The relative expression of the immunohistochemical marker (Figure 2m,q,u) showed that cells in the 50% group collectively expressed higher levels of KRT18, 5, and 14 with 0.7, 0.8, and 0.6 folds respectively compared to the 100% group (*p*-value = 1.73 × 10^−9^, 0.004 and 1.76 × 10^−9^ respectively). The 0% group comprising solely of an adipose-derived stem cell line showed the lowest expression levels of KRT18, 5, and 14 with fold changes of 0.1, 0.01, and 0.4, respectively (*p*-value = 2.95 × 10^−16^; 1.46221 × 10^−12^; and 1.21 × 10^−3^; respectively) compared to the 50% group.

### 2.2. 378 miRNAs Differentially Expressed between Primary Keratinocytes and ADSCs

miRNAs represent an important vehicle for communication between cells. To evaluate the differentially expressed miRNA, GeneChip^®^ miRNA arrays were used to profile the expression in primary human keratinocytes and ADSCs. A total of 378 miRNAs were identified as differentially expressed miRNAs (DEmiRNAs) between primary keratinocytes and ADSCs (Figure 3a,b; Appendix A and Appendix A). Of these, 114 miRNAs (30.16%) were upregulated in keratinocytes while 264 miRNAs (69.84%) were downregulated. Microarray data was validated with qPCR analysis for the selected miRNA targets using individual miRNA assays. The following miRNAs, miR-30b-5p (3.1 fold, *p*-value = 0.002), miR-30c-5p (2.5 fold, *p*-value = 0.002), and miR-203a (5018 fold, *p*-value = 0.03), followed the same pattern of significant differential expression and showed upregulation in keratinocytes. The expression of miR-34a-3p (0.1 fold, *p*-value = 0.008) showed a significant downregulation in keratinocytes in a similar pattern to that in the microarray analysis. Furthermore, miR-29b-3p (0.6 fold, *p*-value = 0.12), miR-195-5p (1.1 fold, *p*-value = 0.48) and miR-374a-5p (0.9 fold, *p*-value = 0.24) were non-significant in both assays. Our validation of selected individual miRNA revealed overall agreement with the microarray data.

### 2.3. Differentially Expressed miRNAs–mRNAs Interactome and Enrichment Analysis of the Upregulated miRNA Targets in Keratinocytes Reveals 14 Pathways

The target genes of the identified significant differentially expressed miRNAs (DEmiRNAs) were investigated separately for upregulated and downregulated miRNAs. A total of 659 unique target mRNAs related to 33 upregulated miRNAs in keratinocytes were identified by combining the three target regions, UTRs (3′ and 5′) and CDS (Appendix A). A similar analysis was performed on the downregulated miRNAs in keratinocytes and a total number of 555 unique target mRNAs for 58 downregulated miRNAs were observed (Appendix A). To investigate the relation between the upregulated miRNAs and their target genes expression in skin, we explored the Expression Atlas database “https://www.ebi.ac.uk/gxa/home (accessed on 2 December 2022)” and collected the tissue related gene expression data from the FANTOM5 [35] dataset with a cut-off score. The cut-off score (CS) calculated as follows:(1)$$CS=x_{i}-\bar{x},\;$$
where $x_{i}$: is the expression value of the gene and $\bar{x}$: is the mean value of total 13,476 genes expressed in Skin, $\bar{x}=\frac{\sum_{i=1}^{n}x_{i}}{N}$.

In this case, a list of 109 unique target genes related to 26 upregulated miRNAs were identified, including AGO2, CDKN1A, MAPK1, MCL1, SEPTIN2, SMAD5, TP53, and TSC1 (Figure 4b; Appendix A). Additionally, 14 pathways were found to be significantly enriched with this list of 109 genes, including signaling by interleukins, RUNX3 regulated CDKN1A transcription, membrane trafficking, vesicle-mediated transport, and others (Figure 4b; Appendix A).

### 2.4. Proteome Profiling and Integrated Analysis with Differentially Expressed miRNAs Highlights miRNA-Mediated Regulations

The protein content in the lysate of primary keratinocytes and ADSCs was experimentally explored, using proteome profiler arrays (Appendix A). Interestingly, only two proteins were significantly upregulated in keratinocytes, which were Epidermal growth factor (EGF) and Interleukin 1-alpha (IL-1α), as shown in Figure 5a.

Integrated analysis between DEmiRNAs and differentially expressed proteins (DEproteins) in the lysate of the studied cell types was conducted. The fisher’s exact test suggested a strong association between a number of DEmiRNAs and the upregulated proteins (*p*-value = 0.04; Appendix A). This integrated analysis presented two predictions for miRNA-mediated gene regulations and their protein products. EGF is likely to be directly controlled by the downregulated miRNAs miR-485-5p and miR-6765-5p or the upregulated miR-4459 (Figure 5b). On the other hand, IL-1α can be controlled by 5 of the upregulated miRNAs, 4 isomers of miR-30 (b, c, d, and e)-5p, and miR-181a-5p (Figure 5c).

## 3. Discussion

The interaction between keratinocytes and stem cells is not only crucial to understand the wound healing process, but also to identify the key regulators of stem cell differentiation into the epidermal lineage. Culturing keratinocytes alongside ADSCs in monolayer allow for cell–cell contact and communication. In our study, a co-culture of keratinocyte and stem cell lines showed genotypic and phenotypic changes allowed by the cross-communication between cells from the two sources. The expected result was the activation of intracellular signaling cascades that enhanced ADSCs differentiation. The effect of direct co-culture on ADSCs differentiation toward other target cell lineages has been previously reported. For example, ADSCs’ potential for osteogenic differentiation was enhanced when co-cultured at a 50:50 ratio with dental pulp stem cells while the effect was diminished with the use of EVs release inhibitor. In support of our findings, this study confirmed the positive effect of co-culture on enhancing cell differentiation, through cell signaling mediators exchanged between the co-cultured cells [36]. Similarly, direct co-culture has also promoted adipogenesis in ADSCs when seeded at a ratio of 70:30 with umbilical vein endothelial cells and cultured in adipogenic differentiation media [37]. Furthermore, the capacity of ADSCs to differentiate into keratinocyte-like cells was described in another co-culture system where keratinocytes were cultured in a transwell above a monolayer of ADSCs. In this model, secreted molecules and growth factors travel through the pores by gravity and induced the differentiation of ADSCs monolayer. In addition, the authors investigated the effect of keratinocyte conditioned medium on ADSCs. Starting at day 7, ADSCs demonstrated gene and protein expression of epidermal markers KRT5, 14, involucrin, filaggrin, and stratifin, comparable to those of keratinocytes. Furthermore, it was not until day 10, when 20–30% of ADSCs changed their morphology from the typical spindle-like appearance of stem cells into a polygonal morphology resembling that of keratinocytes [38].

In our direct co-culture model, upregulation of p63 expression was detected after 7 days to a level resembling that of HaCaT. This marker can be considered as a key transcription factor for the epidermal lineage, being the first gene product distinguishing epidermal progenitor cells, as well as a prevalently expressed marker in proliferating keratinocytes [39,40,41]. The upregulation of p63 suggests that the ASC52 in the co-culture started the commitment of differentiation into epidermal-like cells. In agreement, a previous study showed nuclear p63 expression at day 7 when cultivating MSCs derived from umbilical cord in keratinocyte-specific media with EGF and calcium [42]. KRT18 is a major component of intermediate filaments that acts as an early epithelial differentiation marker, expressed exclusively in simple epithelium prior to stratification [40,43]. Our gene and protein expression data showed the upregulation of KRT18 in the co-cultured cells in a similar trend to p63 gene expression, which supported the evidence for early epidermal differentiation. KRT18 has been shown in differentiating MSCs induced by a cocktail of growth factors, including KGF, EGF, HGF, and IGF-2 for 14 days and altered their fibroblastic morphology to epithelial-like [44]. As none of these differentiation inducers were added to the culture, the intracellular communication was expected to trigger the same effect. Furthermore, KRT5 and 14 were upregulated in our direct co-culture on both the gene expression and protein levels. Dos Santos et al. (2019) reported that the expression of KRT14 in umbilical cord MSCs cultured in keratinocyte media reached its peak at the first day of cultivation followed by abrupt descend at day 4 and maintained a steady state until day 14 [42]. This could be attributed to the role of KRT14 in sustaining proliferation in mitotically active basal keratinocytes followed by downregulation when cells become committed to differentiation [45]. Undifferentiated ADSCs in monoculture showed expression of the epithelial basal marker KRT14 at the protein level, in agreement with previous studies [46,47]. KRT1 and KRT10 are epidermal stratification markers expressed by differentiated keratinocytes in the suprabasal cutaneous layers, including the stratum corneum [5,41]. Longer differentiation protocol in literature showed upregulation of KRT10 starting at day 11, which could explain the absence of upregulation in our 7-day culture [42]. A temporal expression analysis of various epidermal differentiation markers in our system could be considered as an interesting future analysis.

Cell-to-cell communication can occur through several approaches, including receptor-mediated events, direct cell–cell contact, and cell–cell synapses. Often released within EVs, miRNAs are one important mediator in communication, and they disseminate through the extracellular fluid to act as signaling molecules by altering gene expression and protein production in the recipient cell. miRNA-mediated cell-cell communication can be achieved through direct exchange of exosomes between adjacent cells, as well as by shuttling exosomes through the systemic circulation [48,49,50]. Additionally, miRNAs have been shown to regulate various aspects of wound healing including cell proliferation, migration, collagen biosynthesis, and vascularization. Moreover, the field of miRNA-based therapeutics is emerging with vast potential to improve wound healing through targeting of antagomir treatments [51].

In keratinocytes, differential miRNA expression showed that miR-203a was among the most highly expressed. The miRNA miR-203a is one of the most abundant miRNA species in the skin and plays a major role in keratinocytes proliferation and differentiation, alongside the miR-30 family [52,53,54]. On the other hand, miR-34a was downregulated. This result was expected as the cells involved in this study were normal healthy keratinocytes, as the overexpression of miR-34a is known to inhibit keratinocyte proliferation and promote apoptosis [55].

Other major mediators of cell–cell communication are cytokines, which are responsible for a wide range of functions across non-immune cells, including a trophic role in the cell repair and regeneration [56,57]. In the context of stem cell differentiation, chemokines mediate vital cellular processes by establishing the cell communication between proliferating and migrating cells [56]. In this study, both cell types are known for their natural ability to produce cytokines. The secretion can aim at modulating the surrounding tissues in physiological conditions or in response to stimuli, such as cellular stresses imposed by infection, inflammation, tissue damage, or specific culture conditions [58,59]. The expression patterns of a group of cytokines and growth factors as a relevant part of the proteome was explored, with a focus on ADSCs differentiation into epidermal-like cells in response to keratinocytes signaling. Out of the 105 studied proteins, EGF and IL-1α were upregulated in keratinocytes. Keratinocytes are known to both produce and respond to EGF, as it is considered as a major regulator for epidermal homeostasis. The production of EGF by keratinocytes was in agreement with our findings [60,61]. Endogenous growth factors play a major role in orchestrating the proliferative phase in epithelization and are essential for effective wound healing [5,62]. In the epidermis, EGF regulates the barrier function, terminal cell differentiation, cell adhesion, protease secretion, and wound healing [63]. Nevertheless, EGF has long been considered as a crucial additive in epidermal cell culture systems as it stimulates keratinocyte migration and proliferation, as well as ADSCs differentiation into the epidermal lineage [5,42]. Adding the cell culture supernatant of HaCaT cells that are induced to overexpress EGF to ADSCs was associated with enhanced proliferation, migration, and invasion of ADSCs. These findings were abolished when HaCaT were transfected with the EGF inhibitor small interfering RNA (siEGF) [64]. Clinically, the topical application of EGF accelerated healing of split-thickness cutaneous wounds through the stimulation of keratinocytes migration across the wound bed [65].

Epidermal keratinocytes, similar to all epithelial cells with a barrier function, are rich in IL-1α in the physiological state. Upon skin injury, trauma, or infection, IL-1α is released promptly, inducing the local inflammation necessary to initiate wound healing [3,66]. Interestingly, IL-1α is amongst the most frequently reported keratinocyte secretion in culture supernatant, which supports our finding. In skin wounds, IL-1 can mobilize locally located stem cells, as well as enhance the keratinocyte migration [58].

The cluster related to miR-30 is known to be functionally involved in cell fate determination and lineage differentiation of mesenchymal stem cells into adipogenic, chondrogenic, and osteogenic lineages [67]. However, to the best of our knowledge, there are no reports of the direct association of this family to epidermal homeostasis or ADSCs differentiation into epidermal-like cells. The miRNA mir-30a has been shown, in systems other than skin, to block the release of inflammatory cytokines, including IL-1α. However, our results showed the upregulation of miR-30b, c, d, and e and not miR-30a. Different members of miR-30 clusters share a common seed sequence near the 5′ end, but they differ in compensatory sequences near the 3′ end, targeting different genes, and pathways [68,69,70]. The positive regulation of miRNA on proteins can be explained by: (1) miRNA-mediated post-transcriptional upregulation, (2) translation upregulation, or (3) competing with repressive proteins, preventing them from binding to their target sites, leading to increased mRNA stability, thus promoting the expression of the target protein [71]. The upregulation of miR-181a was found to deaccelerate keratinocyte proliferation and promote keratinocyte differentiation when induced by high calcium or UVA irradiation [72,73]. On the other hand, miR-4459 plays a role in decreasing the stemness of human embryonic stem cells through inhibiting its target proteins Cell Division Cycle Protein 20 Homolog B (CDC20B) and Autophagy-Related Protein 13 (ATG13) [74]. To the best of our knowledge, there was no evidence pointing in the direction of miR-4459 mediated EGF regulation in the literature.

On the other hand, miR-485 downregulation in human skin has been previously reported, specifically in terminally differentiated keratinocytes [53]; however, the miR-485 mediated EGF interaction in the context of epidermal development or stem cell differentiation has not been reported before, to the best of our knowledge. KRT17 is known to be a direct target of miR-485, while the signaling cascade involving miR-485/KRT17 may result in suppressing EGFR in oral squamous cell carcinoma cell lines [75]. The accumulated evidence, including the miR485-EGF inhibitory regulation shown here, postulate that miR-485 may constitute an appealing target to be investigated in MSCs differentiation into epidermal-like cells. The inhibitory regulation between downregulated miR-6765 and upregulated EGF is another predicted miRNA-target interaction, which should be explored. To the best of our knowledge, this interaction has not been reported in the context of epidermal development, skin repair, or stem cell differentiation.

In summary, this study reported a direct co-culture model that can be used to study the cell-to-cell interaction in monolayer, including stem cell differentiation to the epidermal lineage. The integrated analysis of miRNA–protein characterization predicted novel pathways for the regulation of EGF and IL-1α in keratinocytes. The investigation of these pathways may help in providing new concepts for stem cell differentiation into epidermal cells as well as for wound repair. Based on our findings and the known role of IL-1α in regulation of cell differentiation, this cytokine should be studied as a media additive for stem cells differentiation into keratinocytes. The limitations of this study included the use of cell lines in the co-culture experiments rather than primary cells. Obtaining enough primary cell numbers to conduct these experiments would be challenging, which is the reason beyond modelling with cell lines instead. The co-culture duration was limited to only 7 days in order to prove the efficiency of the system and to detect early differentiation. Extension of the culture could help in showing more positive cells for the studied markers as well as the expression of late keratinocyte markers. Another limitation for analyzing the study finding was the vast possibilities of intermediate effectors, potential feedback loops, and the post-translational changes affecting the predicted miRNA-mediated gene regulations. Future studies should focus on the experimental validation of the newly proposed miRNA-mediated gene and protein regulations to provide a better understanding of their associated signaling cascades. Loss/gain function studies for the effect of suggest-miRNA to EGF and IL-1α can provide biological evidence for our computational model. Additionally, miRNA-based approach can be explored for in vitro differentiation of stem cells into epidermal cell lineage, as well as in non-healing in vivo wound models. The application of EGF, IL-1α, or a combination of them may be investigated for therapeutic potential.

## 4. Materials and Methods

### 4.1. Epidermal Differentiation Induction in Direct Co-Culture Model

HaCaT cell line (accession: CVCL_U602, Cellosaurus database) of spontaneously transformed, non-tumorigenic keratinocytes isolated from histologically normal skin (Elabscience Biotechnology Inc., Houston, TX, USA) and ASC52 (accession: CVCL_0038, Cellosaurus database) and hTERT immortalized adipose-derived Mesenchymal stem cells (ATCC, Manassas, VA, USA) were used in this study. Both cell lines were cultured in DMEM (Gibco, Billings, MT, USA) with 10% fetal bovine serum (FBS; Life Technologies, São Paulo, Brazil) and 10,000 units penicillin and 10 mg streptomycin/mL (Sigma-Aldrich, St. Louis, MO, USA). Upon reaching 90% confluence, the cells were dissociated with Trypsin-EDTA (Sigma Aldrich, St. Louis, MO, USA), stained with 0.4% trypan blue (1:1) and counted using a TC20 automated cell counter (Bio-Rad Inc., Singapore). Cells from both cell lines were mixed with a concentration of 1:1 in a direct co-culture system. Control groups were either HaCaT alone as a positive control or ASC52 alone as a negative control. The groups were designated as 0, 50, & 100% representing the percentage of HaCaT in the mixture. Cells were seeded at a density of 4 × 10^3^ cell/cm^2^ and kept at 37 °C and 5% CO_2_ for 7 days, with media change every second day. On day 7, cells were either fixed for immunocytochemistry or harvested for RNA extraction and gene expression analysis.

### 4.2. Primary Cell Isolation

Full thickness skin biopsies were obtained from healthy donors during abdominoplasty and/or breast reduction procedures under the ethical approval no. 2015/177-31 by the Swedish Ethical Review Authority. The fat portion was carefully separated. Skin cut into 2–3 mm^2^ then incubated in 1:1 volume of 10 mg/mL Dispase II solution (Gibco, Tokyo, Japan) overnight at 4 °C. The epidermis was then gently peeled from the dermis and incubated with Trypsin-EDTA (Sigma-Aldrich, St. Louis, MO, USA) on a tube rotator at 37 °C for 30 min. Trypsin was deactivated with media with 10% FBS. The cell suspension was allowed to pass through a 70 µm cell strainer (Corning, New York, NY, USA), and then, the keratinocyte suspension was centrifuged at 700 RCF for 4 min. The cell pellets were washed twice with phosphate buffered saline (PBS) (Life Technologies, Grand Island, NY, USA) before cells were resuspended in keratinocyte serum free media (Life Technologies, Grand Island, NY, USA) supplemented with bovine pituitary extract and epidermal growth factor (Life Technologies, Grand Island, NY, USA), and the media was changed every other day. The fat tissue was cut into 0.5–1 cm^2^ slices. Collagenase I (1mg/mL) (Life Technologies, Grand Island, NY, USA) was then added at a 3:1 ratio of the tissue volume and incubated on a tube rotator at 37 °C for 90 min. The tissue solution was then centrifuged at 700 RCF for 4 min, the oil phase was removed, and pre-warmed DMEM with 10% FBS was added to stop the enzymatic digestion. Digested tissue was then passed through a 70 µm cell strainer and washed twice with serum-free DMEM. ASDCs were re-suspended in DMEM with 10% FBS and 1% penicillin-streptomycin and seeded in monolayer culture.

### 4.3. Human miRNome Profiling with Microarrays

Total RNA, including small RNAs, were extracted from primary keratinocytes and ADSCs using miRNeasy kit (Qiagen, Hilden, Germany). Briefly, cell pellets were lysed using QIAzol lysis reagent with mechanical agitation. Cell lysates were dissolved in chloroform and the aqueous phase supernatant was loaded into RNeasy mini column, washed and eluted in RNase free water. RNA yield was measured using NanoDrop 1000 spectrophotometer (Thermo Fisher Scientific, Waltham, MA, USA).

Low molecular weight RNA molecules were labelled with the FlashTag Biotin RNA Labeling Kit (Affymetrix, Santa Clara, CA, USA). Briefly, 500 ng RNA from each sample was ligated to a poly (A) tail followed by binding to a biotinylated signal molecule. miRNA microarray hybridization was then performed with Affymetrix GeneChip miRNA Array 3.0 (Affymetrix, Santa Clara, CA, USA), according to manufacturer’s instructions. Briefly, biotin-labeled samples were incubated with hybridization master mix at 99 °C for 5 min, followed by 45 °C for another 5 min. Hybridization was performed in a rotating hybridization oven (60 rpm) at 48 °C for 18 h. The arrays were washed and stained on GeneChip automated fluidics station and scanned with an Affymetrix GCS 3000 7G-plus scanner (Affymetrix, Santa Clara, CA, USA). The microarray data were analysed using the Transcriptome Analysis Console (TAC)^®^ (Thermo Fisher Scientific, Waltham, MA, USA). Following quality check (Appendix A), differential expression (DE) analysis was performed between the two study groups for homo sapiens specific miRNA with ID contain “hsa-miR” and log2 fold change ≥2 with *p*-value < 0.05 (Figure 1a,b). DE result was annotated using the Affymetrix GeneChip^®^ miRNA 4.0 array annotation, version HG38.

### 4.4. Gene and miRNA Expression

Quantitative real-time PCR was performed to detect the expression of early and late epidermal markers from 5 independent cell line replicates of direct co-culture (n = 5). Total cellular RNA was extracted using RNeasy mini kit (Qiagen, Hilden, Germany). RNA was reverse transcribed into cDNA using Maxima First Strand cDNA Synthesis Kit (Thermo Fisher Scientific, Waltham, MA, USA) following the elimination of double-stranded DNA as recommended by the manufacturer. Gene expression was determined by the PowerUp© SYBR green master mix (Applied biosystem, Waltham, MA, USA). Sequences for the oligonucleotide primers of target genes are listed in Appendix A.

For miRNA quantification in keratinocytes and ADSCs, total RNA, including small RNAs, as described above were reverse transcribed with the miScript II or miRCURY LNA Reverse-Transcription kits (Qiagen, Hilden, Germany) with RNA input of 250 ng or 10 ng respectively, according to the manufacturer’s instructions. The miScript SYBR^®^ Green or the miRCURY^®^ LNA SYBR^®^ Green were used according to the availability of the marker of interest in either workflow (Appendix A) following the manufacturer’s protocols. Gene and miRNA expression levels were quantified in a 7500 Fast Real-Time PCR System (Applied Biosystem, Thermo Fisher) and the assays were performed in a minimum of three technical replicates. Gene and miRNA expression were normalized against the endogenous controls Glyceraldehyde 3-phosphate Dehydrogenase (GAPDH) or U6 snRNA, respectively and the fold change was calculated with the 2^−ΔΔCT^ method [76].

### 4.5. DEmiRNAs-mRNA Interaction Network and Pathway Enrichment Analysis

To identify the target genes of the significant DEmiRNAs, the list of up and downregulated miRNAs were uploaded separately to the miRwalk database v3 “http://mirwalk.umm.uni-heidelberg.de (accessed on 1 December 2022)” [77]. The targeted genes were determined according to the following criteria, (1) the targeted mRNAs should be present on the three databases—TargetScan [78], miRDB [79] and miRTarBase [80], (2) the miRwalk score should be ≥0.95. In order to obtain the “Skin” related expression from the upregulated DEmiRNA-targeted mRNAs in keratinocytes, an expression dataset from the FANTOM5 project in the Expression Atlas [35] was collected. A mean cut-off score was applied on the dataset to draw genes which were highly expressed in skin tissues. The resulting DEmiRNAs-mRNA interactome diagram was generated using the Cytoscape software (v3.8.2) [81]. The Reactome database v83 “https://reactome.org (accessed on 2 December 2022)” [82] was considered to identify the enriched pathways from the up- and downregulated DEmiRNA-targeted mRNAs.

### 4.6. Immunocytochemistry Staining

Immunocytochemistry (ICC) was used to detect the expression of the key epidermal differentiation markers (KRT5, KRT14, and KRT18). Following cell fixation in ice-cold methanol 99% for 15 min at −20 °C, endogenous peroxidase activity was quenched with H_2_O_2_ (10 min). Cells were then permeabilized by incubation for 15 min at room temperature with 0.05% Triton X-100 (Thermo Scientifc, IL, USA). Non-specific binding was blocked using 3% bovine serum albumin in PBS, for 1 h. Primary antibodies, listed in Appendix A, were incubated overnight with the fixed cells at 4 °C followed by incubation with the biotinylated mouse and rabbit specific secondary antibody for an hour at room temperature. The immune complex was visualized using the streptavidin-biotin immunoenzymatic antigen detection system where the streptavidin-enzyme conjugate binds to the biotin present on the secondary antibody. Positive cells were stained with chromogen 3-Amino-9-Ethylcarbazole (AEC) using detection immunohistochemistry kit (Abcam, Cambridge, UK) following the manufacturer’s instruction. Then the cells were rinsed with Tris-buffered saline (TBS) and counterstained with 8GX alcian blue solution (Sigma-Aldrich, St. Louis, MO, USA) for 2 min before a final wash and mounting with anhydrous mounting medium. For negative controls, PBS was applied instead of the primary antibody. ICC were run in triplicate for each individual antibody stain (n = 3). Stained cells photographed under inverted microscope (CKX53, Olympus Corp., Tokyo, Japan) using the Imageview software version X64 (Olympus Corp., Japan). To quantify stain intensity, colour deconvolution was performed using ImageJ 1.53c with the necessary plugins (Wayne Rasband National institutes of health, Bethesda, MD, USA) then setting a unified threshold for integrated pixel density.

### 4.7. Proteome Profiling

Keratinocytes and ADSCs from 3 donors were harvested and solubilized in lysis buffer 17, according to the manufacture instructions (R&D Systems Inc., Minneapolis, MN, USA). Total protein concentration was determined using the DC protein assay (Bio-Rad Inc., Hercules, CA, USA), following the manufacturer’s instructions. Proteome Profiler™ Human XL Cytokine Array (R&D Systems, Inc., Minneapolis, MN, USA) was used to simultaneously assess soluble human proteins and their differential expression between the two cell types. Following the manufacturer’s instructions, the array membranes were blocked for 1 h on a rocking platform, and 150 µg of the cell lysates were incubated with array membranes overnight at 4 °C. The arrays were then incubated with a cocktail of biotinylated detection antibodies for 1 h followed by chemiluminescent detection with Streptavidin-HRP. Membranes were imaged using the ChemiDoc™ MP imaging system (Bio-Rad Inc., Hercules, CA, USA). The pixel densities at each capture spot were quantified and normalized to the reference spots of each blot. Images were analyzed using ImageJ 1.53c (Wayne Rasband National institutes of health, MD, USA) where the mean intensity corresponded to the relative expression of each blotted protein in the cell lysate.

### 4.8. miRNA-Mediated Gene and Protein Regulations (Integrated Analysis)

Integrated analysis was performed between DEmiRNAs and DEproteins in keratinocytes and ADSCs with the assumption that the resulting protein products were directly affected by miRNA-target interaction. To maintain a reliable resource for miRNA target genes, only experimentally validated targets from the miRwalk database were curated and cross-checked with the miRTarBase as well as the miRNA-gene interactions annotated in the Affymetrix GeneChip^®^ miRNA 4.0 array annotation, version HG38. All possible alternative nomenclature was considered for each gene and protein and were used to search for possible miRNA-gene interactions on the aforementioned databases.

### 4.9. Statistical Analysis

For PCR-based assays and ICC, the data were analysed using the Data Analysis ToolPak (Microsoft^®^ Excel, Microsoft^®^ Office 365, Redmond, DC, USA), and the graphs were created using GraphPad Prism Version 9 (GraphPad Software Inc., San Diego, CA, USA). Statistical significance was evaluated using Student’s *t*-test for unequal variance. Bar charts showed the mathematical mean and the standard error of mean. To estimate the DEmiRNAs from the microarray analysis, *p*-value < 0.05 was considered as the level of significance. To identify the enriched pathways in the analysis, Benjamin-Hochberg (BH) corrected *p*-value < 0.05 was applied. Fisher’s exact test was used to test a null-hypothesis, assuming there was no regulation between DEmiRNAs and the genes associated with DEproteins.

## Figures and Tables

**Figure 1 ijms-24-04956-f001:**
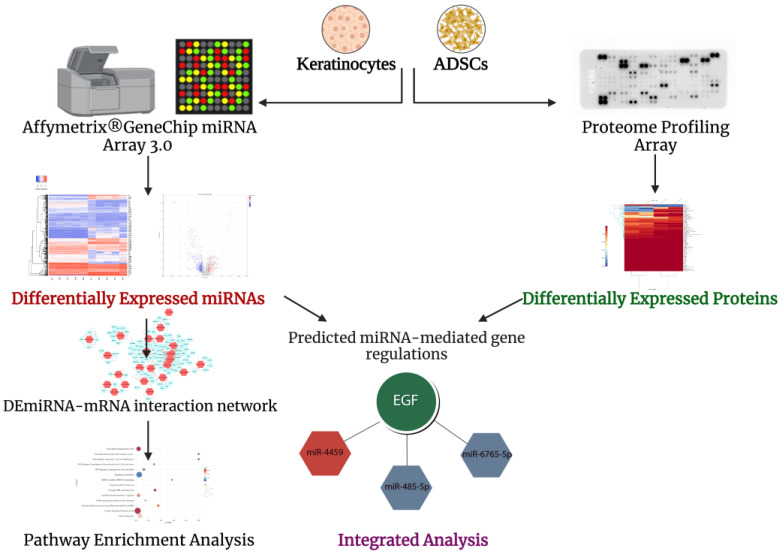
Overview of the computational and experimental analysis of the primary keratinocytes and ADSCs. This figure was created in BioRender “https://biorender.com/ (accessed on 8 January 2023)”.

**Figure 2 ijms-24-04956-f002:**
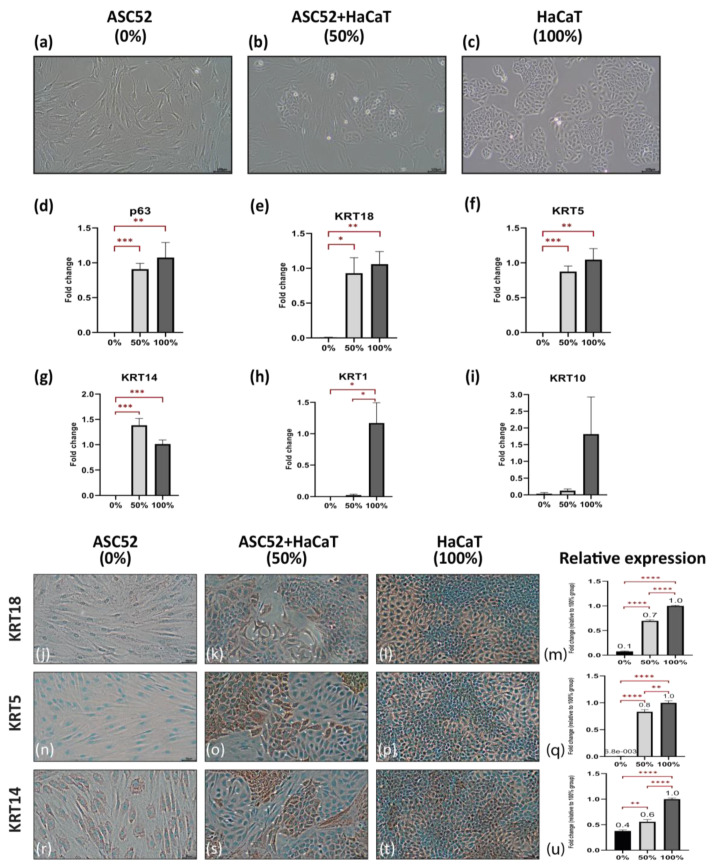
Live images shows the distinct morphology of (**a**) spindle shape ASC52 (0%), (**b**) the combination of ASC52 with HaCaT in co-culture (50% group) and (**c**) polygonal ‘‘cobblestone’’ shape of HaCaT (100% group). Scale bars = 100 µm. Images are representatives of 3 independent experiments. Gene expression of (**d**) p63, (**e**) Keratin (KRT) 18, (**f**) KRT 5, (**g**) KRT 14, (**h**) KRT 1 and (**i**) KRT 10 is shown as the mean of fold change of five independent experiments with reference to the 100% group and the standard error of mean. Immunocytochemistry characterization for KRT18 (**j**–**l**), KRT5 (**n**–**p**) and KRT 14 (**r**–**t**) showed upregulation of these markers in the co-culture (50%) group, especially in ASC52 cells (spindle shape cells), in close proximity to HaCaT colonies. Scale bars = 50 µm. Image representative of 3 independent experiments and—at least—9 read-outs were analyzed for (**m**) KRT 18, (**q**) KRT 5 and (**u**) KRT 14. * *p* ≤ 0.05, ** *p* ≤ 0.01, *** *p* ≤ 0.001 & **** *p* ≤ 0.0001.

**Figure 3 ijms-24-04956-f003:**
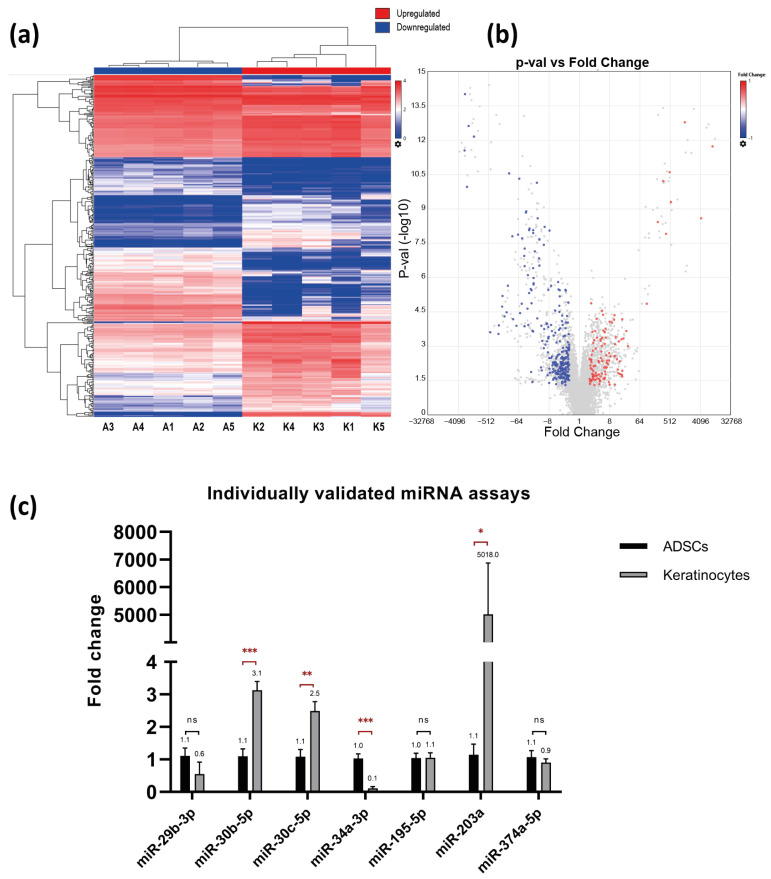
(**a**) Heatmap diagram of the absolute expression levels of miRNA in each sample of the keratinocyte group (K) and ADSCs group (A). (**b**) Volcano plot shows the clustering of upregulated (red) and downregulated (blue) miRNAs. The insignificant differentially expressed miRNAs represented in grey color. (**c**) Validation of selected miRNA expression using individual miRNA PCR assays. Data represented as the mean of fold change of five independent experiments (n = 5) and the standard error of mean. In agreement with microarray analysis, there was upregulation of the expression of miR-30b-5p, mir-30c-5p and miR-203a-3p, while miR-34a-3p was downregulated in keratinocytes. In addition, no significant difference was detected for miR-29b-3p, miR-195-5p or miR-374a-5p.* *p* ≤ 0.05, ** *p* ≤ 0.01 & *** *p* ≤ 0.001.

**Figure 4 ijms-24-04956-f004:**
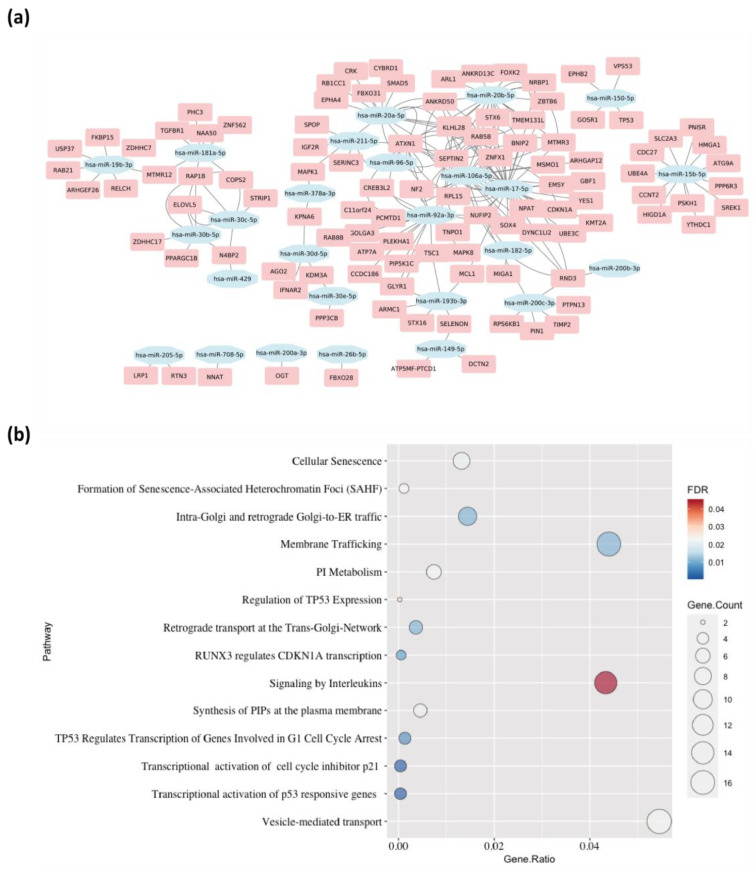
(**a**) miRNA-mRNA interaction network represents 334 targeted genes related to 23 upregulated miRNAs. (**b**) The upregulated miRNAs and their target genes expression can affect 14 signaling pathways in the skin.

**Figure 5 ijms-24-04956-f005:**
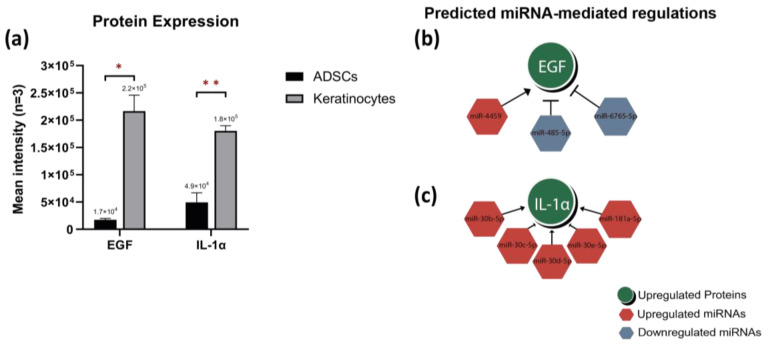
(**a**) The differential protein analysis showed higher expression level of EGF and IL-1α in keratinocytes. Both EGF (**b**) and IL-1α (**c**) can be linked to miRNA regulation through integration analysis. * *p* ≤ 0.05, ** *p* ≤ 0.01.

## Data Availability

The data presented in this study are available in the Appendix A, as well as on request from the corresponding author.

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
