# Peer review of "miRNome and Proteome Profiling of Human Keratinocytes and Adipose Derived Stem Cells Proposed miRNA-Mediated Regulations of Epidermal Growth Factor and Interleukin 1-Alpha"

_ijms, 2023, doi:10.3390/ijms24054956_

Round 1

Reviewer 1 Report

Comments and Suggestions for Authors

The review paper entitled, “miRNome and proteome profiling of human keratinocytes and adipose derived stem cells proposed miRNA-mediated regulations of epidermal growth factor and interleukin 1-alpha” described the interaction between human keratinocytes and adipose-derived stem cells (ADSCs) in promoting differentiation of ADSCs toward the epidermal lineage.  Overall, this manuscript is well written. However, there are some minor points the authors should revise for clarification and publication.

1.     On day 7, cells were either fixed for immunocytochemistry or harvested for RNA extraction and gene expression analysis. How to justify and further confirm that ADSCs have been differentiated to epidermal and not due to the original HaCaT in the co-culture?

2.     For primary cell isolation,  what is the collagenase ratio 3:1 means? Is it collagenase to tissue? Please clarify. What is the passage used for the primary keratinocytes that are isolated from the skin tissues for testing?

3.     There are some formatting errors and writing errors in this manuscript which need to be carefully checked and corrected, eg: 2-3 mm2 should be mm2 (line 418), 37°C (line 420), agitation (line 439), representatives (line 125), space between number and °C and etc.

4.     The font size for figure 1,3 (x-, and y- axis), 4 and 5 are too small. Suggest increasing figures, bar graphs (Fig 3c) and font sizes.

5.     The figure legend should follow the sequence. Figure 2, should explain b first followed by c and not c then b.

6.     Fold change was calculated with the 2-ΔΔCT method. Please cite the method used.

7.     What is the novelty of this study and its future application? 

Author Response

Dear Reviewer 1,

Thanks a lot for taking the time to provide us with your constructive observations. We took all your notes into consideration and here is below our answer for your questions.

  1. On day 7, cells were either fixed for immunocytochemistry or harvested for RNA extraction and gene expression analysis. How to justify and further confirm that ADSCs have been differentiated to epidermal and not due to the original HaCaT in the co-culture?

We can confirm the differentiation based on the following findings

  1. The cells that showed positivity in the immunocytochemistry are those adjacent to the HaCaT colonies.
  2. The relative expression of the studied markers by gene expression or immunocytochemistry are expected to be around 50% as most of these markers have negligible expression in ASC52, while HaCaT expression was set to 100%. The fact that the gene expression was very close to HaCaT for P63, KRT18, KRT5 and KRT 14 can suggest differentiation of the ASC52. The immunocytochemistry findings had the same trend of the gene expression, especially that the protein expression is expected to follow the gene upregulation. These findings, in addition to the immunocytochemistry pattern of expression, strongly suggest ASC52 differentiation into the epidermal lineage from our perspective.

  1. For primary cell isolation, what is the collagenase ratio 3:1 means? Is it collagenase to tissue? Please clarify. What is the passage used for the primary keratinocytes that are isolated from the skin tissues for testing?

Yes, it is collagenase to the fat tissue volume ratio. The text was modified for clarity. Primary cells were maximally used at passage 3.

  1. There are some formatting errors and writing errors in this manuscript which need to be carefully checked and corrected, eg: 2-3 mm2 should be mm2 (line 418), 37°C (line 420), agitation (line 439), representatives (line 125), space between number and °C and etc.

Thank you for picking these errors up. All of them were corrected and the manuscript was double checked for similar errors.

  1. The font size for figure 1,3 (x-, and y- axis), 4 and 5 are too small. Suggest increasing figures, bar graphs (Fig 3c) and font sizes.

The figures were updated according to the reviewer suggestion.

  1. The figure legend should follow the sequence. Figure 2, should explain b first followed by c and not c then b.

Corrected according to the reviewer’s note.

  1. Fold change was calculated with the 2-ΔΔCT method. Please cite the method used.

Citation was added according to the reviewer’s suggestion.

  1. What is the novelty of this study and its future application?

A summary paragraph was added after the discussion to highlight the significance of the study as well as future studies and possible applications.

Reviewer 2 Report

In this manuscript, the authors analyzed miRNome and proteome profiling of human keratinocytes and adipose derived stem cells (ADSCs). They found that miRNAs are differentially expressed between keratinocytes and ADSCs, and it is implied that some miRNAs might regulate EGF and IL-1α expression. Although miRNome and proteome results are informative, most of data are descriptive and preliminary. In addition, since HaCaT cells are different from primary normal epidermal keratinocytes in terms of gene and protein expression, results based on HaCaT cells have limited information. It seems that additional experiments are needed.

Major comments

1. The authors used HaCaT cells for co-culture, but this cell line have different gene and protein expression pattern compared to normal epidermal keratinocytes freshly isolated from skin. Thus, the authors should use primary cells for co-culture. Or, at least, they must explain the limitation of their analysis and avoid overstatement.

2. In the abstract section, please explain the aim of this research more clearly. For what purpose the authors needs to analyze interaction between keratinocytes and adipose-derived stem cells (ADSCs) and differentiation of ADSCs? Also, the conclusion is incomplete and looks like just a description of result. Please clearly explain the significance, importance, or impact of your findings in your research field.

3. Link between co-culture experiment and miRNA profiling is not explained well.

4. Page 3, line 117 “Direct co-culture enhances epidermal differentiation of ADSCs” seems to be overstatement, since the author just analyzed the expression of several KRTs. They should check other parameters as well to confirm epidermal differentiation.

5. Insufficient results are provided to explain miRNA-mediated regulation of EGF and IL-1α. They need to perform gain and loss-of-function experiments of miRNAs which may interact with EGF and IL-1α.

6. It is odd that IL-1α is upregulated in keratinocytes but miR-30s are also upregulated in keratinocytes. Because miR-30 targets IL-1α, upregulation of miR-30 should cause downregulation of IL-1α. Do the authors have any explanation regarding this discrepancy?

7. Discussion section is redundant and it is difficult to read out important part of their discussion. Also, the authors should discuss adequacy of their results by comparing with other publications, not just introduce other papers.

Minor comments

1. Font size of Fig. 1, 4, and 5 should be larger, otherwise it is difficult to read small characters.

2. “HaCat” should be “HaCaT” (“T” is capital letter).

3. Resolution of Fig. 2(a)-(c) is poor. It should be replaced to photos with higher resolution.

Author Response

Dear Reviewer 2,

Thank you for provide us with your constructive notions. We took all your points into consideration and here is below our answer for your questions.

Major comments

  1. The authors used HaCaT cells for co-culture, but this cell line have different gene and protein expression pattern compared to normal epidermal keratinocytes freshly isolated from skin. Thus, the authors should use primary cells for co-culture. Or, at least, they must explain the limitation of their analysis and avoid overstatement.

We agree with the reviewer regarding the presence of differences between HaCaT and primary keratinocytes, however HaCat represents the best available keratinocyte cell model. Technically, it would be difficult to obtain enough cell number of primary cells to perform the co-culture studies. A statement about this limitation was added to the manuscript.

  1. In the abstract section, please explain the aim of this research more clearly. For what purpose the authors needs to analyze interaction between keratinocytes and adipose-derived stem cells (ADSCs) and differentiation of ADSCs? Also, the conclusion is incomplete and looks like just a description of result. Please clearly explain the significance, importance, or impact of your findings in your research field.

In response to the reviewer notes, the aim of the study was clarified in the abstract. A summary paragraph was added after the discussion to highlight the significance of the study.

  1. Link between co-culture experiment and miRNA profiling is not explained well.

A statement was added to section 2.2, before discussing the miRNA microarray results to establish the link, based on the reviewer’s comment.

  1. Page 3, line 117 “Direct co-culture enhances epidermal differentiation of ADSCs” seems to be overstatement, since the author just analyzed the expression of several KRTs. They should check other parameters as well to confirm epidermal differentiation.

The sentence has been changed in the manuscript, according to the reviewer concern. Kindly note that the study duration is 7 days only. The expression of P63 and this combination of KRTs, within this timeframe, was very interesting and suggests early differentiation. The results are confirmed by gene as well as protein expression. Nevertheless, a statement was added to the limitations to address the short culture duration.

  1. Insufficient results are provided to explain miRNA-mediated regulation of EGF and IL-1α. They need to perform gain and loss-of-function experiments of miRNAs which may interact with EGF and IL-1α.

We mentioned in the manuscript that experimental validation is needed as future work direction. Now, another statement was added to highlight the reviewer’s concern.      

  1. It is odd that IL-1α is upregulated in keratinocytes but miR-30s are also upregulated in keratinocytes. Because miR-30 targets IL-1α, upregulation of miR-30 should cause downregulation of IL-1α. Do the authors have any explanation regarding this discrepancy?

We would like to thank the reviewer for highlighting this important point. We added our explanation to the text, in order to avoid confusion of the readers. As you can see in text, miR-30a-5p is the most studied member of the miRNA-30 family and overexpression blocks the release of IL-1α. This effect was not studied in keratinocytes, to the best of our knowledge. Furthermore, we reported the upregulation of miR-30b, c, d and e and not miR-30a. In addition, we added a note in the limitation section regarding the experimental evidence of this relation.

  1. Discussion section is redundant and it is difficult to read out important part of their discussion. Also, the authors should discuss adequacy of their results by comparing with other publications, not just introduce other papers.

The discussion was revised and rewritten according to the reviewer’s note. Comparison with previous work was reported wherever possible. The references were updated and reorganized according to the updated text in different sections.

Minor comments

  1. Font size of Fig. 1, 4, and 5 should be larger, otherwise it is difficult to read small characters.

The figures were updated according to the reviewer’s comment

  1. “HaCat” should be “HaCaT” (“T” is capital letter).

The manuscript was revised and all the occurrences with ‘t’ was corrected to ‘T’. We apologize for this mistake.

  1. Resolution of Fig. 2(a)-(c) is poor. It should be replaced to photos with higher resolution.

The figure was replaced. Kindly, consider that during pdf compilation, some figures can be distorted. The original figures are submitted as single files with higher resolution.

Round 2

Reviewer 2 Report

The authors have tried to answer the concerns which I listed in the previous review report.

Although additional experiments are preferable to raise quality of this manuscript, those might be performed in the future studies and I don't have any more comments.